# Use of Leaves as Bioindicator to Assess Air Pollution Based on Composite Proxy Measure (APTI), Dust Amount and Elemental Concentration of Metals

**DOI:** 10.3390/plants9121743

**Published:** 2020-12-09

**Authors:** Vanda Éva Molnár, Dávid Tőzsér, Szilárd Szabó, Béla Tóthmérész, Edina Simon

**Affiliations:** 1Department of Physical Geography and Geoinformatics, University of Debrecen, H-4032 Debrecen, Hungary; molnarvandaeva@gmail.com (V.É.M.); szabo.szilard@science.unideb.hu (S.S.); 2Department of Ecology, University of Debrecen, H-4032 Debrecen, Hungary; tozser.david@windowslive.com; 3MTA-DE Biodiversity and Ecosystem Services Research Group, H-4032 Debrecen, Hungary; tothmerb@gmail.com

**Keywords:** urban pollution, foliage dust, chlorophyll, biomonitoring, bioassessment, ecological status, multimetric index

## Abstract

Monitoring air pollution and environmental health are crucial to ensure viable cities. We assessed the usefulness of the Air Pollution Tolerance Index (APTI) as a composite index of environmental health. Fine and coarse dust amount and elemental concentrations of *Celtis occidentalis* and *Tilia × europaea* leaves were measured in June and September at three sampling sites (urban, industrial, and rural) in Debrecen city (Hungary) to assess the usefulness of APTI. The correlation between APTI values and dust amount and elemental concentrations was also studied. Fine dust, total chlorophyll, and elemental concentrations were the most sensitive indicators of pollution. Based on the high chlorophyll and low elemental concentration of tree leaves, the rural site was the least disturbed by anthropogenic activities, as expected. We demonstrated that fine and coarse dust amount and elemental concentrations of urban tree leaves are especially useful for urban air quality monitoring. Correlations between APTI and other measured parameters were also found. Both *C. occidentalis* and *T. europaea* were sensitive to air pollution based on their APTI values. Thus, the APTI of tree leaves is an especially useful proxy measure of air pollution, as well as environmental health.

## 1. Introduction

Air pollution is a growing problem worldwide, and it is especially serious in large cities. With the speed of urbanisation and industrialisation in the past decades, pollution emitting sources have increased both in number and intensity. Urban growth is expected to continue; therefore, large-scale urban planning and development projects are urgent challenges [1].

Among air pollutants, the issue of particulate matter deserves particular attention because of its adverse health effects [2]. There are numerous natural sources of dust pollution, but dust originating from anthropogenic activities is of major concern due to the heavy metals usually found in these particles [3]. Heavy metals accumulate in the human body, and they are toxic even at a relatively low level [4]. Heavy metals also pose an ecological risk to plants, such as trees in urban habitats. While certain metals are essential for normal plant functions, others can have an adverse effect on biochemical and physiological parameters [5], seed germination, plant growth, and morphological parameters [6,7]. There is considerable uptake from the soil via the root system, but certain metals have significant bioavailability from atmospheric deposition through aboveground vegetative organs [8,9].

It is especially important that urban green spaces effectively reduce particulate pollutants [10,11]. Trees provide an alternative way of monitoring urban air quality due to their constant exposure and cost-effective sampling. The Air Pollution Tolerance Index (APTI) was proposed by Singh et al. [12] as an index to express the capacity of plants to counter the adverse effects of air pollution. APTI is a composite measure of tree-health; it is calculated from the ascorbic acid content of leaves, total chlorophyll content, pH of leaf extract, and relative water content. These parameters are affected by air pollutants, making APTI an especially useful tool to assess air pollution [13,14]. Plants with APTI categories of excellent, very good and good performers are suggested for the development of green belts [15]; APTI can also give information about the pollution level of the cities based on plants. Evidently, APTI is a compositional index or multimetric index. These kinds of indices are statistical tools that amalgamate many different measures, metrics and/or indices to create a representation of overall quality or performance. Composite indices are used for assessing the ecological quality status of various kinds of ecosystems, biological and ecological resources. They synthesize different kinds of data, deriving a single index that reflects the overall effects of human influence [16]. Based on APTI, the most tolerant and best-suited plants to protect against the pollution can be selected, and the sensitive plant species can also be used as bioindicators [17].

The aim of this study was to assess the level of air pollution based on the parameters of tree leaves under increasing anthropogenic impacts. We also assessed the usefulness of the Air Pollution Tolerance Index (APTI) as a simple and useful compositional index of environmental health. Urban, rural, and industrial areas were studied using the leaves of *C. occidentalis* (Linné) and *T. europaea* (Linné). Dust deposition, APTI, and elemental concentration were measured and statistically analysed. Our hypotheses were: (i) the load of deposited dust and elemental concentration on tree leaves would increase at the urban study site compared to the industrial and rural sites, (ii) there would be a correlation between APTI values and other measured parameters, such as the amount of fine and coarse dust and the elemental concentration in leaf tissues, and (iii) the studied species (*C. occidentalis* and *T. europaea)* would be sensitive to air pollution, and (iv) the APTI values of leaves would reflect the air quality as a composite proxy measure to assess the level of air pollution in the cities.

## 2. Results

### 2.1. Dust Amount

Differences in fine and coarse dust depositions were measured at the study sites with various pollution loads (urban, industrial, rural area), during the months of June and September, using the species *C. occidentalis* and *T. europaea*. There were significant differences in the amount of fine and coarse dust among the study sites based on *C. occidentalis* (fine dust: F = 26.260, *p* = 0.001; coarse dust: F = 24.763, *p* < 0.001). We only found significant difference in coarse dust on the *T. europaea* leaves (F = 4.827, *p* = 0.043). There were no significant differences in fine and coarse dust of *C. occidentalis* leaves between the studied months (*p* > 0.05). There was a significant difference between the months in fine dust deposited on *T. europaea* leaves (*t* = 2.880, *p* = 0.016) (Appendix A). There was no significant difference in the amount of dust on the leaf surfaces of *C. occidentalis* and *T. europaea* (fine dust: *t* = −0.247, *p* = 0.805; coarse dust *t* = 0.638, *p* = 0.525).

### 2.2. APTI and Elemental Concentration

Differences in the means of APTI values and its parameters (ascorbic acid, total chlorophyll content, pH of leaf extract, and relative water content) were tested among study sites (urban, industrial, rural), months (June, September), and species (*C. occidentalis*, *T. europaea*).

There was a significant difference along the urbanisation gradient based on *T. europaea* leaves for relative water content (F = 9.602, *p* = 0.002), the pH of the leaf (F = 3.760, *p* = 0.048), the APTI (F = 4.310, *p* = 0.033), and concentrations of aluminium (F = 60.953, *p* < 0.001), barium (F = 9.894, *p* = 0.013), chromium (F = 37.080, *p* < 0.001), copper (F = 7.719, *p* = 0.022), iron (F = 88.037, *p* <0.001), potassium (F = 5.240, *p* = 0.048), manganese (F = 7.968, *p* = 0.020), sodium (F = 14.096, *p* = 0.005) and strontium (F = 13.148, *p* = 0.006) (Appendix A). The total chlorophyll (F = 4.350, *p* = 0.004) and concentrations of barium (F = 15.512, *p* = 0.007), calcium (F = 16.670, *p* = 0.006), chromium (F = 6.576, *p* = 0.040), magnesium (F = 19.331, *p* = 0.004), manganese (F = 6.067, *p* = 0.046), sodium (F = 11.178, *p* = 0.014) and strontium (F = 26.545 *p* = 0.002) of *C. occidentalis* leaves also differed significantly among sites. The parameters for *T. europaea* leaves differed between July and September, except for total chlorophyll (Appendix A). All parameters of *C. occidentalis* differed between months, except the concentrations of ascorbic acid (*p* > 0.05) (Appendix A). The two species showed differences in many of the studied parameters in a uniform trend. Ascorbic acid content (*p* < 0.001), leaf pH (*p* < 0.001), relative water content (*p* < 0.05), and APTI (*p* < 0.001) itself were significantly higher in *C. occidentalis* than *T. europaea*. *C. occidentalis* scored an average APTI of 12.9 with a minimum of 8.4 and a maximum of 16.4. The average APTI of *T. europaea* was 8.7, ranging from 6.3 to 10.7 across all sites.

### 2.3. Separation of Study Sites Based on Deposited Dust and APTI Parameters

Canonical Discriminant Analysis resulted in two discriminant functions (DFs): DF1 and DF2 explained 59.0% and 41.0% of the total variance, respectively, for *C. occidentalis* (Table 1). We found similar results in the case of *T. europaea*, where the DF1 and DF2 explained 62.7% and 37.3% of the total variance, respectively. The significance of the Chi-square statistics suggested that DF1 and DF2 had significant discriminatory abilities in *C. occidentalis* (*p* < 0.05), while in *T. europaea*, only the DF1 showed a significant difference (*p* < 0.001). Plots of the discriminant scores of the two discriminant functions demonstrated a clear separation of the study sites (Figure 1) for each species.

The DF1 correlated positively with the total chlorophyll content (*r* = 0.379) and relative water content (r = 0.051), while a negative correlation was found with the ascorbic acid content of *C. occidentalis* (r = −0.101). The DF2 correlated positively with fine dust (r = 0.153) and leaf pH (*r* = 0.036), while there was a negative correlation with coarse dust (r = −0.297) for *C. occidentalis*. There was a positive correlation between DF1 and relative water content of *T. europaea* (r = 0.792), while the other parameters correlated with DF2 with the coarse dust (r = −0.596), leaf pH (r = −0.495), total chlorophyll content (r = 0.345), ascorbic acid content (r = −0.224), and fine dust (r = −0.061).

### 2.4. Separation of Study Sites Based on the Elemental Concentrations of Leaf Tissue

Two functions were generated as a result of CDA, and both functions had significant discriminatory abilities (*p* < 0.001; Table 2), except in the case of *T. europaea*. The DF1 and DF2 explained 95.3% and 4.7% of the total variance, respectively, for *C. occidentalis*. Furthermore, the DF1 explained 99.6%, and the DF2 explained only 0.4% of the total variance for *T. europaea*.

For *C. occidentalis*, the DF1 correlated with the concentrations of magnesium (*r* = 0.261), lead (r = 0.173), copper (*r* = 0.169), zinc (*r* = 0.127), nickel (*r* = 0.118), manganese (*r* = 0.091), iron (*r* = 0.072), cadmium (*r* = 0.068), aluminium (*r* = 0.063), barium (*r* = 0.063), potassium (*r* = 0.057), and cobalt (*r* = 0.037). The DF2 correlated positively with the concentrations of strontium (*r* = 0.093), calcium (*r* = 0.075), and sodium (*r* = 0.065), while a negative correlation was found with chromium (*r* = −0.050). All elemental concentrations correlated with the DF1 for *T. europaea* (Table 2). Based on the discriminant scores of the functions, the groups were clearly separable in the case of each species (Figure 2).

### 2.5. Correlation between APTI and the Measured Environmental Parameters

We found significant correlations (r_s_) between the concentrations of manganese and coarse dust amount; between the concentrations of manganese, and fine dust amount; and between the concentrations of chromium and manganese, and APTI (Appendix A) in *T. europaea* leaves. For *C. occidentalis* leaves, there was a significant correlation between the concentrations of calcium, chromium, magnesium and strontium, and coarse dust amount; between the concentrations ofcalcium, copper, manganese and sodiumand fine dust amount; and between the concentrations of barium, cadmiumsodium and APTI (Appendix A).

## 3. Discussion

We demonstrated that monitoring the amount of deposited dust on the surface of urban tree leaves can be an especially useful and effective method for monitoring urban air quality. Leaves act as dust traps due to specific factors in the anatomy of tissues, like trichome and stomata density. In this study, we found an increased amount of fine dust deposited on tree leaves at the urban study area compared to the industrial and rural sites. Furthermore, this suggests that the presence of increased vehicular traffic had a notable effect on the dust emission at the urban site. The urban site was located in the city center where pollutants easily accumulate due to unfavorable diffusion conditions and the presence of emission sources like heavy traffic. Apart from primary sources, secondary sources, caused by constant resuspension of road dust by vehicles, also have a large influence. Similarly, in the city of Gandhinagar, India, Chaudhary [18] found increased dust depositions on tree leaves in traffic zones compared to commercial and residential zones. However, the highest values were still found in an industrial zone. Based on our findings, there was no additional dust pollution at the industrial site compared to the rural site. The industrial site was at the peri-urban area of the city, the vehicular emission was moderate, and the diffusion of pollutants was not hindered by tall rows of buildings like in the city center. Varga et al. [19] studied the pollution sources using fossil carbon (i.e., ^14^C) analysis of tree and grass leaves: urban sites are more exposed to traffic-related pollutants, while in the suburbs, the main source is wood burning (heating). The largest fossil carbon level was found at the crowded crossroads in a densely populated area [20].

Comparing our findings on dust deposition in June and in September, we found that there was an increase in the amount of both fine and coarse particles in September. The early autumn ploughing of neighboring agricultural areas in September have contributed to the increase in dust deposition. However, there was no significant difference in the amount of dust on the leaves of *C. occidentalis* and *T. europaea*. Dust deposition on the leaf surface depends on numerous morphological factors. Earlier studies showed that the dust capturing ability could be similar for certain species, while for others, it can differ significantly [21,22].

The chlorophyll content relates to the productivity in plant tissue. Certain pollutants can cause depletion of photosynthetic pigments, such as chlorophylls; thus, they can inhibit photosynthetic activity [23]. In our study, the total chlorophyll content of tree leaves was significantly higher at the rural site compared to the urban and industrial sites. The reduction at these latter sites is presumably caused by the presence of air pollutants. For instance, fine dust deposition, as discussed above, was elevated at the urban site. Regarding the industrial site, a reduction in chlorophyll content can also be caused by other pollutants that were not measured in our study. Earlier studies also presented results of decreased photosynthetic pigments in plants in polluted areas [18,24,25]. However, other authors found increased photosynthetic pigments with elevated pollution load [26]. These findings could also be explained by the compensatory mechanism of plants against pollution stress, i.e., the chlorophyll content increases with pollution level until a critical level, then it decreases [27]. Based on our findings, the total chlorophyll content in leaves also showed depletion over time from June to September. This is a natural progression in deciduous trees nearing the end of the growing season.

We found higher ascorbic acid content, leaf pH, and relative water content in *C. occidentalis* leaves than in *T. europaea* leaves. Higher values of these parameters generally indicate a higher tolerance for plants. Ascorbic acid protects the plant from oxidative compounds and is essential for several physiological mechanisms. The relative water content helps maintain physiological balance through increased transpiration rates, while the leaf pH influences the stomatal permeability of pollutants [28,29,30]. Along with these parameters, the APTI was also higher in *C. occidentalis* leaves than in *T. europaea* leaves. This clearly indicates that the tolerance level of *C. occidentalis* is elevated against air pollution; both *C. occidentalis* and *T. europaea* are categorized as sensitive based on their average APTI values according to the categorization of Singh et al. [12]. The APTI values, however, did not differ among the study areas.

The study areas could be completely separated based on the concentrations of aluminium, barium, calcium, chromium, copper, iron, potassium, manganese, sodium and strontium in leaf tissue. The concentrations of chromium, iron and strontium were lower at the rural site than at the industrial site. Similar findings were reported for several elements by other researchers when comparing the elemental composition of plant tissue at polluted and unpolluted sites [22,31,32]. Trees can easily accumulate metals, especially in leaf tissue; therefore, they have often been recommended for biomonitoring purposes.

Several researchers found a correlation between the dust load and the reduction in chlorophyll content in plant foliage as a result of increased dust pollution [33]. Our results show a correlation between APTI values and fine dust amount in the case of *T. europaea*. In our study, there was a correlation between APTI values and the elemental concentrations of chromium and manganese and the APTI values based on the measured parameters of *T. europaea*. For *C. occidentalis,* there was a significant correlation between barium, calcium, and sodium concentration and APTI values. Nadgórska-Socha et al. [5] found a negative correlation between APTI values and iron concentrations in *M. album*, and a positive correlation between copper concentrations and APTI values. Furthermore, they found a positive correlation between the manganese concentration and APTI values for *R. pseudoacacia* leaves. Karmakar and Padhy [17] also found a correlation between APTI values and the concentrations of copper and iron in *A.*
*auriculiformis* leaves, between cadmium and chromium concentrations and APTI values in *E.*
*globulus* leaves, and between the concentrations of copper, zinc, and iron and APTI values in *A.*
*indica* leaves. Based on our correlation results and the results of earlier studies, APTI can be considered a more suitable index for ecophysiological assessments than individual indicators, since plants show different responses to different pollutants [5,34]. In the case of APTI, our results showed that the sensitivity and response of plants are affected by pollutants.

## 4. Materials and Methods

### 4.1. Study Areas and Sample Collection

The study areas were in Debrecen city, the second-largest city in Hungary. It is on the terrain of the Great Hungarian Plain, at 120 m above sea level [35]. The region is exposed to particulate pollution. It is a sink for aerosols arriving with the prevailing northwestern wind from neighboring regions [36,37]. Aerosols from the Sahara often influence air quality as well [38,39]. Three sampling sites were chosen to represent the varying intensities of anthropogenic activities (urban, industrial, and rural; Figure 3). The urban site was in the city center with a high intensity of vehicular traffic. The industrial site was located in the western part of the city, near the Industrial Park. The rural area was in the northern part of the city, close to the border of the Nagyerdő Nature Conservation Area. Tree leaves from the common hackberry (*Celtis occidentalis*) and common lime (*Tilia × europaea*) were collected. There were 3 sampling stations (urban, industrial and rural). In each sampling station, 3 specimens of 2 tree species (*C. occidentalis* and *T. europaea*) were sampled. During the sampling procedure, 10 healthy leaves were collected from each tree specimen. These leaves were pooled before the chemical analysis. There were 2 sampling periods (June and September) for dust and APTI. Thus, altogether 3 stations * 2 species * 3 specimens * 2 sampling periods = 36 samples were collected for dust and APTI. For the elemental analysis, leaf samples were collected only in September; thus, altogether 3 stations * 2 species * 3 specimens * 1 sampling period = 18 samples were collected.

### 4.2. Analysis of Dust Samples

The area of the leaf samples was determined by scanning the leaves in black and white. During the chemical analyses, the leaves were put into a 500 mL plastic boxes. Then, 250 mL of deionized water was added to the leaves. The samples were shaken for 10 min on an orbital shaker. This suspension was filtered through a 150 μm sieve. Then the leaves were shaken in 50 mL deionized water again, repeating the previous procedure [21,31]. This 300 mL of suspension was filtered through two types of filter paper using a vacuum filter machine (N 811 KN.18 Laboport, Pune, India). First, a filter paper of a retention diameter of 5–8 µm was used (Munktell 392, Ahlstrom, Helsinki, Finland) so that the amount of coarse dust could be measured. Then, the filtrate was filtered again using a filter paper of a retention diameter of 2–3 µm (Munktell 391, Ahlstrom, Helsinki, Finland). The amount of fine dust was measured using the gravimetric method; filter papers were weighed before and after filtration to determine the amount of dust collected on the paper. The amount of dust was given in µg cm^−2^, as in mass per area of the leaf surface.

### 4.3. Air Pollution Tolerance Index

APTI values were calculated based on the ascorbic acid content in mg g^−1^ (A), total chlorophyll content in mg g^−1^ (T), pH of leaf extract (P), and relative water content (R) of the tree leaves. Using these parameters, we applied the equation proposed by Singh et al. [12] for APTI:APTI = [A × (T + P) + R]/10(1)

The components of the APTI were measured as follows. The ascorbic acid content was measured with the redox titration method where 2 g of leaf tissue was crushed and homogenized in deionized water. After filtration, the samples were titrated using an iodine solution with starch as an indicator.

Chlorophyll was extracted from approximately 20 mg of fresh leaf tissue using 5 mL of 96% ethanol. The absorbances of the extracts were measured on the wavelengths of 653, 666, and 750 nm using spectrophotometric analysis. Total chlorophyll content (T) was calculated as follows:T (mg g^−1^) = (17.12 × E666 − 8.68 × E653) × V/m × 1000(2)
where V is the volume (mL) of leaf extract, m is the fresh weight (g) of the leaf sample, and E666 and E653 are the absorbances at 666 nm and 653 nm minus the absorbance at 750 nm, respectively. For the pH measurement, 2 g of leaf tissue was crushed and homogenized in 100 mL deionized water. The leaf pH of this extract was measured using a digital pH meter. To determine the relative water content, the fresh weight of individual leaves (FW) was measured. Then, leaves were immersed in water overnight before being weighed again to determine the turgid weight (TW). Finally, leaves were dried in an oven at 70 °C to measure the dry weight (DW). The relative water content (R) was calculated as follows:R (%) = (FW − DW)/(TW − DW) × 100(3)

### 4.4. Analysis of the Elemental Concentration of Leaves

Leaf samples from June that were previously washed were dried at 60 °C and then homogenized with an electric mixer. For the elemental analysis 0.2 g of the samples was digested in 5 mL 65% (*m/m*) nitric acid and 1 mL 30% hydrogen peroxide using a microwave digestion unit (Milestone 1200 Mega, Shelton, CO, USA). Digested samples were diluted to 25 mL with deionized water for the analysis [31]. Inductively coupled plasma optical emission spectrometry (ICP-OES 5110 Agilent Technologies, Santa Clara, CA, USA) was used for the elemental analysis. Duck weed (BCR670, Merck, Ltd.) Certified Reference Material (CRM) was used and the recoveries were within 10% of the certified values for the elements. However, in the CRM, only the indicative values for Cr, Cu, Pb, and Zn were given. The six-point calibration was applied to ensure high precision measurements. Calibration curve was calculated using a multi-element calibration solution (Merck ICP multi-element standard solution IV). The concentrations of aluminium, barium, calcium, chromium, cobalt, copper, iron, potassium, magnesium, manganese, sodium, nickel, lead, strontium and zinc were measured in the leaf samples.

### 4.5. Statistical Analysis

The normal distribution was tested with the Shapiro-Wilk test. The homogeneity of variances was tested with the Levene’s test. The differences between samples were tested using analysis of variance (ANOVA) for each variable. When group variances were unequal, the Games-Howell method was used for pairwise comparison between the groups. For variables, ANOVA was used to compare the means. In the case of months and species, the two-sample *t*-test was used as these categories only had two groups. Canonical discriminant analysis (CDA) was used to reduce dimensions and to identify those variables which most efficiently discriminated the study areas as the dependent variable. Fine dust, coarse dust, total chlorophyll content, ascorbic acid content, leaf pH, and relative water content were used as independent variables. We repeated the CDA with the elemental concentrations (aluminium, barium, calcium, chromium, cobalt, copper, iron, potassium, magnesium, manganese, sodium, nickel, lead, strontium, and zinc) of leaf tissues as independent variables to separate the study areas as dependent variables. We reported the properties of Discriminant Functions (DFs) and their correlations with the independent variables’ observed values (r). Spearman correlation (r_s_) was used to study the correlation between the concentrations of elements and dust amount and APTI values. Statistical analyses were conducted with the SPSS Statistics 20 (IBM Company, Armonk, NY, USA) statistical software [40].

## 5. Conclusions

In this study, the deposited dust amount and elemental concentration of *C. occidentalis* and *T. europaea* were measured at urban, rural, and industrial sites in Debrecen city, Hungary. Fine dust, total chlorophyll, and elemental concentrations contributed the most to the separation of study areas. Heavy vehicular traffic caused increased dust deposition at the urban site in the city centre. The rural site was the least disturbed by anthropogenic activities based on the high chlorophyll and low elemental concentrations in the tree leaves. Our findings demonstrated that roadside trees reduced air pollution by retaining dust particles on leaf surfaces, and they can indicate the pollution levels by altered biochemical parameters. Therefore, tree leaves can be used as reliable bioindicators of urban air pollution. Correlations between dust amount and elemental concentrations and APTI values confirmed that APTI efficiently indicated the level of air pollution. Both *C. occidentalis* and *T. europaea* were sensitive based on their average APTI values. Thus, the APTI values can be used to select pollution tolerant species used for urban greening and/or selecting species for green belt development. Plant species with lower APTI values are especially useful as bioindicators of air pollution and as a proxy for urban health.

## Figures and Tables

**Figure 1 plants-09-01743-f001:**
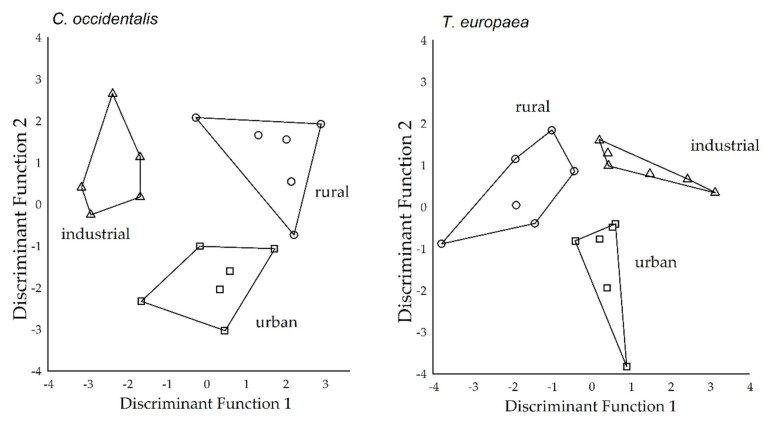
Discriminant score plots of the study sites (where rural, urban, and industrial groups represented the urbanisation gradient) based on dust amount (*N* = 3 pooled samples from one area), ascorbic acid content, total chlorophyll content, pH of leaf extract, and relative water content of tree leaves (N = 3 pooled samples from one area) by species.

**Figure 2 plants-09-01743-f002:**
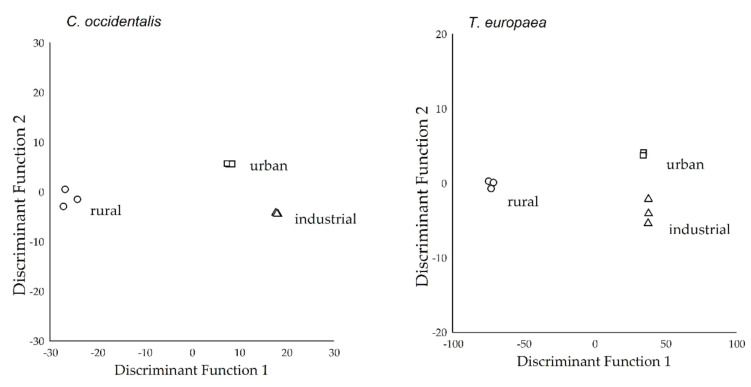
Discriminant score plots of the study sites (where rural, urban, and industrial groups represented the urbanisation gradient) based on the elemental concentrations in leaf tissue (*N* = 3 pooled samples from one area) of the studied species (*C. occidentalis*, and *T. europaea*).

**Figure 3 plants-09-01743-f003:**
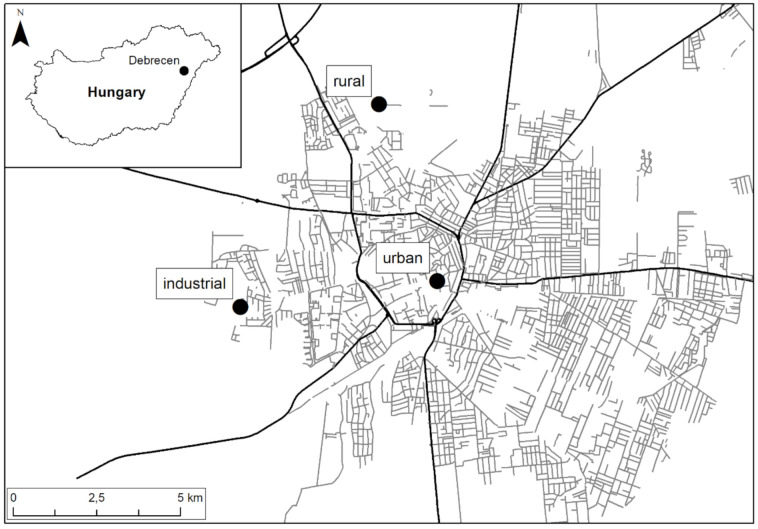
Sampling sites in Debrecen city, Hungary.

**Table 1 plants-09-01743-t001:** The output of Canonical Discriminant Analysis (CDA) of study sites based on dust and APTI parameters.

Function	*C. occidentalis*		*T. europaea*	
	DF1	DF2	DF1	DF2
Eigenvalue	3.3	2.3	2.0	1.2
Percentage of Variance	59.0	41.0	62.7	37.3
Cumulative percentage	59.0	100.0	62.7	100.0
Canonical Correlation	0.9	0.8	0.8	0.7
Wilks’ Lambda	0.1	0.3	0.2	0.5
Chi-square	30.4	13.7	23.5	9.8
df	12.	5	12	5
Significance	0.002	0.018	0.023	0.081

**Table 2 plants-09-01743-t002:** The output of the CDA of study sites based on the elemental concentrations in leaf tissue of the studied species.

Function	*C. occidentalis*		*T. europaea*	
	DF1	DF2	DF1	DF2
Eigenvalue	532.6	26.3	3978.5	15.5
Percentage of Variance	95.3	4.7	99.6	0.4
Cumulative percentage	95.3	100.0	99.6	100.0
Canonical Correlation	1.0	0.9	1.0	0.9
Wilks’ Lambda	0.0	0.04	0.0	0.1
Chi-square	33.6	11.6	38.8	9.8
df	12	5	12	5
Significance	0.001	0.041	<0.001	0.081

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
