# Peer review of "Use of Leaves as Bioindicator to Assess Air Pollution Based on Composite Proxy Measure (APTI), Dust Amount and Elemental Concentration of Metals"

_plants, 2020, doi:10.3390/plants9121743_

Round 1

Reviewer 1 Report

The article discusses monitoring of air pollution by monitoring different parameters of leaves of two species. The authors suggest that Air Pollution Tolerance Index that can be used as a proxy measure of air pollution.

You cannot use the term dust content, since it appears at the leaf surface.  I presume that heavy metals were measured in leaf tissue and dust at the leaf surface. Be consistent.

Title

The title should be more informative e.g.

Introduction

Use the full name of plants (and authors names) when first mentioned.

The amount of ascorbic acid content of leaves, total chlorophyll content, pH of leaf extract, and relative water content are parameters that may change significantly within on species, therefore they show a current condition of tree, but one cannot generally declare certain species as tolerant only based on these parameters.

Hypotheses should be more clearly presented and better supported by “the state of art” (references).

Results

Results should be presented separately for each species.

Methods

APTI is summing apples and pears (APTI = [ A × (T + P) + R] / 10). There is no logic in it at all. I would avoid the use of APTI and I propose separate elaboration of measured parameters that will give an insight into vulnerability of plants at different locations. I think that the relation between toxic metals and dust is especially interesting.

Figure captions should give full information about the graph. Please extend the text.

Discussion

Should be better supported by relevant measurements.

Author Response

Review 1

Comments and Suggestions for Authors

The article discusses monitoring of air pollution by monitoring different parameters of leaves of two species. The authors suggest that Air Pollution Tolerance Index that can be used as a proxy measure of air pollution.

You cannot use the term dust content, since it appears at the leaf surface.  I presume that heavy metals were measured in leaf tissue and dust at the leaf surface. Be consistent.

ANSWER: Corrected. We changed to dust concentration in everywhere.

Title

The title should be more informative e.g.

ANSWER: Modified.

Introduction

Use the full name of plants (and authors names) when first mentioned.

ANSWER: Corrected.

The amount of ascorbic acid content of leaves, total chlorophyll content, pH of leaf extract, and relative water content are parameters that may change significantly within on species, therefore they show a current condition of tree, but one cannot generally declare certain species as tolerant only based on these parameters.

Hypotheses should be more clearly presented and better supported by “the state of art” (references).

ANWER: We evidently agree that these parameters may change within a specimen. But it is also evident that the health condition of a tree specimen is very different depending on the pollution level of a city or district of the city. Thus, using a typical leaf of the tree specimen we get reliably report of the condition of the tree specimen. Hypotheses are more clearly presented as recommended.

Results

Results should be presented separately for each species.

ANWER: As requested, we presented our results separately for each species.

Methods

APTI is summing apples and pears (APTI = [ A × (T + P) + R] / 10). There is no logic in it at all. I would avoid the use of APTI and I propose separate elaboration of measured parameters that will give an insight into vulnerability of plants at different locations. I think that the relation between toxic metals and dust is especially interesting.

ANSWER: Thank you. It is really grateful that you think especially interesting the relation between toxic metals and dust in our study. Urbanization is a leading process worldwide; thus it is especially important to propose and evaluate simple measure of city-health for decision making. It is indeed a hard task. This motivated our study evaluating the usefulness of APTI as such measure.

Figure captions should give full information about the graph. Please extend the text.

ANSWER: Corrected.

Discussion

Should be better supported by relevant measurements.

ANSWER: We added further arguments, as proposed.

Reviewer 2 Report

The title refers to a physiological index (APTI) while in the paper the mineral composition is also discussed (not heavy metals, lines 14,18,34,35,36,53, 102,.....) and it appears that the quantity of particulated material is a good indicator for air quality (lines 117-118). That measures (physiological indicators or trace elements) on plants or parts of plants (leaves and tree bark) are good indicators of anthropogenic pressure on the environment is nothing new. Perhaps the correlation between APTI and mineral composition can be considered a new result but this is not clear from the discussion (and from the title).
Some comments:
1) It is not clear from the paper which results are used in discussion. Are 9 samples (lines 189 and 194) obtained for each species? Sampling done 2 times. Regarding the physiological measures, the two species are different (lines 81-85) but in Figure 3 do they come together? 12 points for each area?
2) The r values in lines 94-96 and lines 107-109 are not clear to me.
3) Pricipal component analysis (PCA) is a more suitable technique for highlighting group or outliers than Canonical discriminant analysis (CDA) which requires the identification of categories.
4) Also for the mineral composition it is not clear how many samples were considered in Figura 4. Some (3) elements have not been considered without any explication.
5) From Figures 3 and 4 it is not clear how APTI is a good indicator, while the mineral composition separates the three zones in an exceptional way. Perhaps the discussion should focus on this difference.
6) Paragraph "4.4. Analysis of heavy metal content" deserves more attention. The six-points calibration was necessary for lack of linerity? No accuracy assessment is reported (analysis of Certified Material) but some of the elements considered (e.g. Al, Ba, Cr) may be quite troublesome (low recoveries). The reference to work #27 does not introduce more details.
7) Lines 59-68. Again it is not clear which data were used for comparison. Differences "among study sites" have been done with data from two sampling period (June, September) and with the two tree species? I.e. with 9 data for each site?

Minor comments:
1) Several of the elements considered are not "heavy metals".
2) Units of measure in the text should be corrected (lines 213, 215, 216, 225).
3) Wavelengths for chlorophyll content measurement is 654 in line 223, but 653 in lines 225 and 227.
4) Line 232. Water content?
5) Line 237 hydrogen peroxide (remove hyphen)
6) In Figures 1 and 2 the letters a and b are used with two different meanings (which figure and statistically significant difference).
7) References in lines 347-350 should be corrected.
8) Bothanical names in italic.

Author Response

Review 2

Comments and Suggestions for Authors

The title refers to a physiological index (APTI) while in the paper the mineral composition is also discussed (not heavy metals, lines 14,18,34,35,36,53, 102,.....) and it appears that the quantity of particulated material is a good indicator for air quality (lines 117-118). That measures (physiological indicators or trace elements) on plants or parts of plants (leaves and tree bark) are good indicators of anthropogenic pressure on the environment is nothing new. Perhaps the correlation between APTI and mineral composition can be considered a new result but this is not clear from the discussion (and from the title).

ANSWER: Thank you for the proposal. We have added it to the manuscript.

Some comments:
1) It is not clear from the paper which results are used in discussion. Are 9 samples (lines 189 and 194) obtained for each species? Sampling done 2 times. Regarding the physiological measures, the two species are different (lines 81-85) but in Figure 3 do they come together? 12 points for each area?

ANSWER: We presented our results separately for each species.

2) The r values in lines 94-96 and lines 107-109 are not clear to me.

ANSWER: Clarified.

3) Principal component analysis (PCA) is a more suitable technique for highlighting group or outliers than Canonical discriminant analysis (CDA) which requires the identification of categories.

ANSWER: We calculated PCA; see  below.

4) Also for the mineral composition it is not clear how many samples were considered in Figura 4. Some (3) elements have not been considered without any explication.

ANSWER: We presented our results separately for each species.

5) From Figures 3 and 4 it is not clear how APTI is a good indicator, while the mineral composition separates the three zones in an exceptional way. Perhaps the discussion should focus on this difference.

ANSWER: Thank you, we included a discussion of it.

6) Paragraph "4.4. Analysis of heavy metal content" deserves more attention. The six-points calibration was necessary for lack of linerity? No accuracy assessment is reported (analysis of Certified Material) but some of the elements considered (e.g. Al, Ba, Cr) may be quite troublesome (low recoveries). The reference to work #27 does not introduce more details.

ANSWER: We completed the elemental content analysis.

7) Lines 59-68. Again it is not clear which data were used for comparison. Differences "among study sites" have been done with data from two sampling period (June, September) and with the two tree species? I.e. with 9 data for each site?

ANSWER: We analysed our results separately for each species. Because of precisity the six-point calibration was used. Aquatic plant CRM was used and the recoveries were within of the 10 % of the certified values for the elements.

Minor comments:

1) Several of the elements considered are not "heavy metals".

ANSWER: Corrected.

2) Units of measure in the text should be corrected (lines 213, 215, 216, 225).

ANSWER: Corrected.

3) Wavelengths for chlorophyll content measurement is 654 in line 223, but 653 in lines 225 and 227.

ANSWER: Corrected.

4) Line 232. Water content?

ANSWER: Corrected.

5) Line 237 hydrogen peroxide (remove hyphen)

ANSWER: Corrected.

6) In Figures 1 and 2 the letters a and b are used with two different meanings (which figure and statistically significant difference).

ANSWER: Figures were modified as requested.

7) References in lines 347-350 should be corrected.

ANSWER: Corrected.

8) Bothanical names in italic.

ANSWER: Corrected.

Round 2

Reviewer 1 Report

As I have already pointed out in the first review,  I do not believe that APTI is composed properly, from the point of view of plant physiology. The parameters included in the index equation have different units, and one cannot sum chlorophyll content (mg/g) and pH (no units) and in addition also water content in %. However, if the authors elaborate single parameters these may be important indicators.

Author Response

Comments and Suggestions for Authors

As I have already pointed out in the first review, I do not believe that APTI is composed properly, from the point of view of plant physiology. The parameters included in the index equation have different units, and one cannot sum chlorophyll content (mg/g) and pH (no units) and in addition also water content in %. However, if the authors elaborate single parameters these may be important indicators.

ANSWER: APTI was proposed in 1983, and has been successfully used in biomonitoring studies ever since. It is a composite index or multimetric index, where the differences in measurement units should not be a problem. A composite index or multimetric index is a statistical tool that amalgamates many different measures, metrics and/or indices to create a representation of overall quality or performance. Composite indices are used for assessing the ecological quality status of various kind of ecosystems, biological and ecological resources. These indices synthesize different kinds of data, sometimes even from multiple levels of biological organization, with the goal of deriving a single index that reflects the overall effects of human influence (Schoolmaster et al 2012). A typical ecological example of the utilization of these kinds of indices is the ecological status assessment of the waterbodies in the European Water Framework Directive. It is usually achieved through multimetric techniques, combining several indices, which address different stressors or different components of the waterbodies (e.g. Hering et al 2006).

Our understanding on the usage of composite/multimetric index instead of single ones that integrating the information of more variables can provide a more general response of the plants to pollution than evaluating individually. Thus, we agree that we lose information based on a given single measure but similarly to a statistical example, means also generalize and we use them in all fields of life. The goal of our study was to assess the usefulness of APTI as a proxy measure of ecosystem health based on air pollution.

References

Donald R. Schoolmaster Jr, James B. Grace and E. William Schweiger 2012: A general theory of multimetric indices and their properties. Methods in Ecology and Evolution 2012, 3, 773–781 doi: 10.1111/j.2041-210X.2012.00200.x

Daniel Hering, Christian K. Feld, Otto Moog & Thomas Ofenböck 2006: Cook book for the development of a Multimetric Index for biological condition of aquatic ecosystems: experiences from the European AQEM and STAR projects and related initiatives.

Hydrobiologia (2006) 566:311–324   DOI 10.1007/s10750-006-0087-2  DOI 10.1007/s10750-006-0087-2

Reviewer 2 Report

The presentation of data has improved (tree species treated separately), but there are several points that need to be explained (e.g. no explication on which samples have been used). I had several difficulties in understanding the answers given in the coverletter, actually in the manuscript there are no the corrections as indicated in the coveretter. The points that I believe need to be modified are (as in my first review):
1) The title continues to refers to APTI only while "Fine dust, total chlorophyll, and elemental concentration were the most to the separation of study sites" (lines 20-21). I think that the paper deals with the use of plant parts (the leaf) as bioindicator. Lines 27-29 and 64-65 are not justified in my opinion, from paragraph 4.5 seems that APTI is not used in data treatment. That APTI show a "gradient" (line 103) is not evident from Table S1.
2) Paragraph 2.1 is not clear. There is no differences in month sampling? So data are treated together? From CDA Figure 1 there are 6 point; i.e. for each tree species we have three sampling point with two replicates (in June and September, the three leafs have been pooled)? Is that correct? Which data have been used for treatment.
3) Looking at table S2, there are important differences in the concentrations of trace elements between June and September, up to 10 times. In Figure 2 only 3 points? Only one sampling? Differences in sampling period should be discussed.
4) Lines 102-106. The F-test is the variance test? But from the text it appears that the values ​​are being compared.
5) The meaning of r values (lines 161-171) is still not clear.
6) In line 242 "The APTI values however did not differ among the study areas", in contrast with end of Abstract (and title, and scope of the paper in introduction).
7) Trace elements concentrations are discussed (good results in Figure 2) but their reliability is not supported by data reported. There is no accuracy assessment as suggested in my previous comments. The method used may be subject to low recoveries for some of the elements considered. I can't find (lines 340-350) details on accuracy even if in the "Author response to report 1:" is stated "Aquatic plant CRM (which one? BCR-060?) was used and the recoveries were within of the 10 % of the certified values for the elements". Why a six-point calibration is not explained: "Because of precisity the six-point calibration was used (???)". How the data (significant figures) are reported in Table S2 is it in agreement with the sensitivity of the analytical method?

Minor comments
Table S3 has wrong heading (S2).
Line 156 Furthermore?
Check "concentration" and "amount". E.g. line 189: amount seems more correct than concentration.

Author Response

Reviewer 2

The presentation of data has improved (tree species treated separately), but there are several points that need to be explained (e.g. no explication on which samples have been used). I had several difficulties in understanding the answers given in the coverletter, actually in the manuscript there are no the corrections as indicated in the coveretter. The points that I believe need to be modified are (as in my first review):

 1) The title continues to refers to APTI only while "Fine dust, total chlorophyll, and elemental concentration were the most to the separation of study sites" (lines 20-21). I think that the paper deals with the use of plant parts (the leaf) as bioindicator.

ANSWER: We modified the title.

Lines 27-29 and 64-65 are not justified in my opinion, from paragraph 4.5 seems that APTI is not used in data treatment. That APTI show a "gradient" (line 103) is not evident from Table S1.

ANSWER: Lines 27-29 and 64-65 are as follows: “The significance of the Chi-square statistics suggested that DF1 and DF2 had significant discriminatory abilities in C. occidentalis (p < 0.05), while in T. europaea, only the DF1 showed a significant difference (p < 0.001).” This means that there were significant different between the treatments according to the Discriminat Analysis.

 2) Paragraph 2.1 is not clear. There is no differences in month sampling? So data are treated together? From CDA Figure 1 there are 6 point; i.e. for each tree species we have three sampling point with two replicates (in June and September, the three leafs have been pooled)? Is that correct? Which data have been used for treatment.

ANSWER: We corrected the section of Results 2.2. from L85-L96, the section of Results 2.5. from L146-L154, the section of Discussion from L205-L222 and the section of Material and Methods 4.1. from L244-253.

 3) Looking at table S2, there are important differences in the concentrations of trace elements between June and September, up to 10 times. In Figure 2 only 3 points? Only one sampling? Differences in sampling period should be discussed.

ANSWER: We corrected the Table S2 because of erratum and wrong data. Accordingly, we corrected the text on the section of Results 2.2. from L85-L96, the section of Results 2.5. from L146-L154, the section of Discussion from L205-L222 and the section of Material and Methods 4.1. from L244-253.

4) Lines 102-106. The F-test is the variance test? But from the text it appears that the values are being compared.

ANSWER: Yes, F-test is used to compare the means based on the variance structure of the samples. Please see Zar, J.H. 2010. Biostatistical Analysis, Pearson, https://www.pearson.com/us/higher-education/program/Zar-Biostatistical-Analysis-5th-Edition/PGM263783.html?tab=contents

5) The meaning of r values (lines 161-171) is still not clear.

ANSWER: “r” values means the correlation between Discriminant Function and measured parameters (i.e. metals). We added a short explanation to the methodology.

6) In line 242 "The APTI values however did not differ among the study areas", in contrast with end of Abstract (and title, and scope of the paper in introduction).

ANSWER: We corrected. Although, APTI values did not differ among the study areas presumably the moderate level of anthropogenic activities, because of the finding correlation between APTI and other measured parameters APTI is an especially useful tool to assess the level of air pollution.

 7) Trace elements concentrations are discussed (good results in Figure 2) but their reliability is not supported by data reported. There is no accuracy assessment as suggested in my previous comments. The method used may be subject to low recoveries for some of the elements considered. I can't find (lines 340-350) details on accuracy even if in the "Author response to report 1:" is stated "Aquatic plant CRM (which one? BCR-060?) was used and the recoveries were within of the 10 % of the certified values for the elements". Why a six-point calibration is not explained: "Because of precisity the six-point calibration was used (???)". How the data (significant figures) are reported in Table S2 is it in agreement with the sensitivity of the analytical method?

ANSWER: We completed the text from L299-L300. During the elemental analysis we checked the limit of detection for each element. When the concentration was below the limit of detection we did not analysed the elements for example in the case of cadmium, which is indicated in the Table S2.

Minor comments

Table S3 has wrong heading (S2).

ANSWER: Corrected.

 Line 156 Furthermore?

ANSWER: Corrected.

 Check "concentration" and "amount". E.g. line 189: amount seems more correct than concentration.

ANSWER: We checked. We corrected and we used the dust amount phrase.

Round 3

Reviewer 1 Report

I still think that the equation for APTI contains data that have very different backgrounds (pH is logarithmic and behaves quite differently than other two) and there is no correction for this effect on the index. I would suggest that authors to use perMANOVA test (permutation MANOVA) (Anderson, 2001) and standardize the data before analysis. This test is based on a distance matrix and can be used with composite data.

Anderson, M.J. 2001. A new method for non-parametric multivariate analysis of variance. Austral Ecology, 26: 32–46.

Author Response

Comments and Suggestions for Authors

I still think that the equation for APTI contains data that have very different backgrounds (pH is logarithmic and behaves quite differently than other two) and there is no correction for this effect on the index. I would suggest that authors to use perMANOVA test (permutation MANOVA) (Anderson, 2001) and standardize the data before analysis. This test is based on a distance matrix and can be used with composite data.

Anderson, M.J. 2001. A new method for non-parametric multivariate analysis of variance. Austral Ecology, 26: 32–46.

ANSWER: We are grateful for the referees for the proposals regarding the data management. We accepted the proposals during the previous revision. We know quite well the perMANOVA method of Marti Anderson. Even Anderson says that this technique is similar to the Discriminance Analysis. Large majority (9 out of 10) of the papers use Discriminance Analysis. The best strategy therefore is to use the Discriminance Analysis. perMANOVA is more flexible, because it is based on distance(s), but it may cause a serious problem when using heterogeneous data set (this is our case). A simple example: for the variable A the range is from 0 to 5; for the variable B the range is from 10 to 10’000; i.e. in the data set there are various kinds of variables. The Manhattan distance (but you can try Euclidean distance) for A is 5, while for the variable B it is 10000-10=9990. This causes a serious problem (bias)! As a solution you may try to standardize the data. But there is no reason to use data manipulation techniques during this analysis. In the Discriminance Analysis there is a canonical standardization. Therefore, we agree that perMANOVA might be useful, but we would like to be conservative and using Discriminance Analysis in the manuscript.

Reviewer 2 Report

There is an improvement in the manuscript but the important issues have not been resolved. ASPI  is frequently cited while in the text are discussed  the parameters that were used to calculate ASPI, it is not the same thing.  "We  assessed the usefulness of the Air Pollution Tolerance Index (APTI) as a composite index of environmental health (lines 13-15)" but it does not seem to me that this is evident from the results presented (see for example Table S1) .

Sampling details have been added but it is not clear what has been done; should be explained better. For each sampling stations (3), for each tree species (2) and for each sampling period (2) have been collected 3 samples on 3 different trees, each composed of 10 healthy leaves. I.e. a total of 36 samples each composed of 10 leaves.  Is this the sampling scheme?  For dust and mineral composition 12 samples? Perhaps it is described correctly and it is my fault having  difficulty understanding, but I still do not find a correspondence between the description of the sampling and the points reported in the two Figures.

My doubts about the ICP OES measures have not been addressed. Moreover none of the elements analysed is certified (line 298) in the CRM used.

Author Response

There is an improvement in the manuscript but the important issues have not been resolved. ASPI is frequently cited while in the text are discussed the parameters that were used to calculate ASPI, it is not the same thing. "We assessed the usefulness of the Air Pollution Tolerance Index (APTI) as a composite index of environmental health (lines 13-15)" but it does not seem to me that this is evident from the results presented (see for example Table S1) .

ANSWER: Thank you so much your helpful recommendations. We corrected the indicated statements in the text from L24-25, L69-71, L332-333.

Sampling details have been added but it is not clear what has been done; should be explained better. For each sampling stations (3), for each tree species (2) and for each sampling period (2) have been collected 3 samples on 3 different trees, each composed of 10 healthy leaves. I.e. a total of 36 samples each composed of 10 leaves.  Is this the sampling scheme?  For dust and mineral composition 12 samples? Perhaps it is described correctly and it is my fault having difficulty understanding, but I still do not find a correspondence between the description of the sampling and the points reported in the two Figures.

There were 3 sampling station (urban, industrial and rural). In each sampling station 3 specimens of 2 tree species (C. occidentalis and T. europaea) were sampled. During the sampling procedure there were collected 10 healthy leaves from each tree specimens. These leaves were pooled before the chemical analysis. There were 2 sampling period (June and September) for dust and APTI. Thus, altogether 3 station * 2 species * 3 specimens * 2 sampling period = 36 samples were collected for dust and APTI. For the elemental analysis leaves samples were collected only in September; thus, altogether 3 station * 2 species * 3 specimens * 1 sampling period = 18 samples were collected. We corrected from L243-250 in the text.

My doubts about the ICP OES measures have not been addressed. Moreover none of the elements analysed is certified (line 298) in the CRM used.

ANSWER: We know that the used CRM does not content the all elements but unfortunately better CRM is not available in the commercial traffic. Please find the inserted table which contents the instrumental conditions for ICP-OES for trace elemental analysis

Round 4

Reviewer 2 Report

The description of the sampling is made clearly and the two graphs are finally understandable.

Many of the answers in the cover letter are not actually reported in the text (lines 24-25 does not seem to me to be a proven statement by the data, line 332 is empty). The table added in the cover letter on ICP OES analyzes does not improve the situation.

Apart from the sampling description I see no improvement.

Lines 23-24: the statement is not clear.

Author Response

Reviewer2

The description of the sampling is made clearly and the two graphs are finally understandable.

ANSWER: Thank you.

Many of the answers in the cover letter are not actually reported in the text (lines 24-25 does not seem to me to be a proven statement by the data, line 332 is empty).

The table added in the cover letter on ICP OES analyzes does not improve the situation.

ANSWER: We corrected the text from L24-25 and L339-340. We showed the all instrumental conditions for ICP-OES for trace elemental analysis. We inserted a column in the Table S2, which contains the calculated detection limit of elements for leaves tissue.

Apart from the sampling description I see no improvement.

Lines 23-24: the statement is not clear.

ANSWER: Corrected.
